# Shockwave Therapy Modulates the Expression of BMP2 for Prevention of Bone and Cartilage Loss in the Lower Limbs of Postmenopausal Osteoporosis Rat Model

**DOI:** 10.3390/biomedicines8120614

**Published:** 2020-12-15

**Authors:** Shan-Ling Hsu, Wen-Yi Chou, Chieh-Cheng Hsu, Jih-Yang Ko, Shun-Wun Jhan, Ching-Jen Wang, Meng-Shiou Lee, Tsai-Chin Hsu, Jai-Hong Cheng

**Affiliations:** 1Center for Shockwave Medicine and Tissue Engineering, Kaohsiung Chang Gung Memorial Hospital, Chang Gung University College of Medicine, Kaohsiung 833, Taiwan; hsishanlin@yahoo.com.tw (S.-L.H.); murraychou@yahoo.com.tw (W.-Y.C.); t1234@cgmh.org.tw (C.-C.H.); kojy@cgmh.org.tw (J.-Y.K.); b9502077@cgmh.org.tw (S.-W.J.); cjwang1211@gmail.com (C.-J.W.); tsaichin1219@gmail.com (T.-C.H.); 2Department of Orthopedic Surgery, Sports Medicine, Kaohsiung Chang Gung Memorial Hospital, Chang Gung University College of Medicine, Kaohsiung 833, Taiwan; 3School of Nursing, Fooyin University, Kaohsiung 831, Taiwan; 4Department of Chinese Pharmaceutical Science and Chinese Medicine Resources, China Medical University, 91, Hsueh-Shih Road, Taichung 406, Taiwan; leemengshiou@mail.cmu.edu.tw; 5Medical Research, Kaohsiung Chang Gung Memorial Hospital, Chang Gung University College of Medicine, Kaohsiung 833, Taiwan; 6Department of Leisure and Sports Management, Cheng Shiu University, Kaohsiung 833, Taiwan

**Keywords:** osteoporosis, shockwave therapy, lower limbs, prevention of bone loss, chondroprotective effect

## Abstract

Osteoporosis (OP) causes bone loss and weakness, increasing the risk of bone fracture. In this study, rats were divided into Sham, OP, SW(F) (0.25 mJ/mm^2^ with 1600 impulses to the left medial femur), and SW(T) (0.25 mJ/mm^2^ with 1600 impulses to the left medial tibia). The bone strength results following SW(T) were better than SW(F) in the modulus, extension at peak load, handleability, and strain at break. SW(T) had the best prevention for bone loss in both lower limbs of ovariectomized (OVX) rats. The cartilage cellular matrixes of both knees were improved in SW(T) and SW(F) compared to that of OP. Serum bone morphogenetic protein 2 (BMP2) in rats undergoing SW(T) or SW(F) was significantly improved compared to that in Sham and OP. The expressions of BMP2, BMP4, and SMAD family member 4 (Smad4) in addition to the Wnt family member 3A (Wnt3a) and Cyclin D1 signaling key factors were significantly induced in the cartilage of both knees by shockwave (SW). SW(T) presented the best efficacy to induce serum BMP2 to prevent bone loss from both lower limbs. Here, we display the protective effects of SW therapy to induce BMP2, BMP4, Smad4, Wnt3a, and Cyclin D1 signaling factors for cartilage loss in both knees of OVX rats.

## 1. Introduction

Osteoporosis (OP) is a common bone disorder that escalates with age, especially in women after menopause [1]. The etiology of OP involves several factors, including estrogen deficiency, drugs, smoking, a poor diet, and infection [2,3]. OP causes a decrease in bone mineral density (BMD), deterioration of bone quality, and micro-architectural fractures of the bone. Many studies have been performed to assess potential approaches to the prevention or possible regression of postmenopausal bone loss, such as pharmacologic therapy, traditional Chinese medicine, physical exercise, mechanical stimulation, and electrical stimulation [4,5,6]. Pharmaceutical approaches include antiresorptive and stimulating agents, whereas nonpharmaceutical approaches include exercise programs and biophysical interventions [7,8]. Shockwave (SW) therapy, vibration, magnetic fields, and low-intensity pulsed ultrasound (LIPUS) are biophysical interventions that provide options for local treatment of OP at common fracture sites. However, the long-range or systemic effects of these methods are unclear and require further assessment. 

SW therapy has exhibited beneficial effects in the treatment of orthopedic disorders, including tendinopathies, tennis elbow, shoulder rotator cuff pain, and nonunion of long-bone fractures, as well as soft tissue disorders over 30 years of study [9,10,11,12,13]. SW induces subperiosteal callus formation through the creation of microfractures on the cortex [14]. Moreover, many studies have demonstrated that SW promotes the expression of growth factors, including vascular endothelial growth factor (VEGF), transforming growth factor β1 (TGF-β1), insulin-like growth factor 1 (IGF-1), and bone morphogenetic proteins (BMPs) [11,13,15,16]. SW has also been shown to improve blood supply and cell proliferation and to result in eventual tissue regeneration [11,17,18]. The osteogenetic effects of SW have also been demonstrated on callus formation in fracture-healing, nonunion, delayed bone–tendon junction repair, and osteoporotic osteoarthritis [13,19,20,21]. This mechanical, noninvasive method could be a useful tool by which to encourage tissue regeneration. 

The mechanism of SW acts through mechanotransduction at the cellular level, and this technique is currently used to treat a wide variety of musculoskeletal disorders [22]. Focused SW has been demonstrated to exert a chondroprotective effect and to induce new bone formation in various animal models in normal bone, fractured bone, bone after osteotomy, and bone defects [13,21,23]. Other studies have revealed mechanisms that induce oxygen radicals and membrane hyperpolarization to promote the expression of growth factors and the activation of osteoprogenitor cells [24,25]. Previous research results further implied that SW can be used in site-specific treatment and can provide the opportunity to treat OP. For example, SW enhanced local BMD by inducing new bone formation in osteoporotic bone in an ovariectomized (OVX) goat model [26]. In 2017, Lama and colleagues demonstrated the beneficial anti-osteoporotic effects of SW alone or in combination with raloxifene treatment, supporting their hypothesis of a systemic effect of SW alone or in combination therapy [27]. In particular, systemic concentrations of osteogenesis-related factors are increased in the serum after focused SW according to the results of our previous study [24]. Therefore, we considered that SW to the left femur or left tibia might stimulate the expression of key factors to promote bone growth and to protect articular cartilage of the knees, not only via a local effect but also through long-range effects in an OVX rat model.

## 2. Materials and Methods 

### 2.1. Animals

The rats were treated humanely in accordance with the Guide for the Care and Use of Laboratory Animals. The Division of Laboratory Animal Resources at Kaohsiung Chang Gung Memorial Hospital (CGMH) provided veterinary care to the rats. All rats were housed at 23 ± 1 °C under a 12-h light and dark cycle and were given food and water. This study was approved by the Institutional Animal Care and Use Committee (IACUC: 2016011101) at CGMH on 1 February, 2016.

### 2.2. Laparotomy and Bilateral Ovariectomy

The female rats were anesthetized with a 1:1 volume mixture of Rompun (5 mg/kg) (xylazine-hydrochloride, Bayer, Leverkusen, Germany) and Zoletil (20 mg/kg) (tiletamine-zolazepam, Virbac, Carros, France). The abdomen of each rat was scrubbed and prepared in a sterile fashion for surgery. A midline incision was made, and laparotomy was performed. Both ovaries of the rats were removed by bilateral ovariectomy [21]. In the laparotomies of the Sham group rats, no ovaries were removed. The abdominal wound was irrigated and closed in a routine fashion.

### 2.3. Focused SW Application

Twenty-four six-week-old Sprague-Dawley ovariectomized (OVX) rats were used in the study. The experimental design is shown in Figure 1. The source of SW was a DUOLITH SD1 (STORZ MEDICAL AG, Tägerwilen, Switzerland). SW was focused on the left medial femur or left tibia at 0.5 cm below the skin surface (Figure 1B,C). The twenty-four animals were randomized into 4 groups: the Sham group did not undergo surgery or SW; the OP group underwent bilateral ovariectomy but did not receive SW, and served as an osteoporotic animal model; the SW(F) group consisted of OVX rats that received 1600 impulses in total of focused SW (0.25 mJ/mm^2^) to each location of the left medial femur one week post-surgery (Figure 1A,B); and the SW(T) group consisted of OVX rats that received 1600 impulses in total of focused SW (0.25 mJ/mm^2^) to each location of the left medial tibia one week post-surgery (Figure 1A,C). All animals were sacrificed at 12 weeks post-treatment. The evaluations performed included pathological analysis, micro-CT scanning, analysis of the bone strength of the femur, and immunohistochemical analysis.

### 2.4. Bone Strength Analysis

The right femur bones were harvested for bone strength analysis via the modulus, extension at peak load, handleability, and strain at break. Whole bone of the right femur was obtained and subjected to bone strength analysis using an MTS machine (MTS, Eden Prairie, MN, USA).

### 2.5. Micro-CT Scanning

Cortical and trabecular bone morphometric parameters were determined in the proximal tibia bone using micro-CT scanning (Skyscan 1076; Skyscan, Luxembourg, Belgium) with an isotropic voxel size of 36 × 36 × 36 μm^3^. The bone mineral density (BMD), trabecular thickness (TbTh), bone volume/tissue volume (BV/TV), and bone porosity were measured and analyzed (Figure 2). The volume of interest (VOI) region of bone morphometry was calculated under 1.75 mm from epiphysis of the left and right tibia and up 1.75 mm from epiphysis of the left and right femur for BMD, TbTh, BV/TV, and bone porosity with a semiautomatic contouring method by Skyscan CT-Analyser Software (Skyscan 1076; Skyscan, Luxembourg, Belgium). The transverse and sagittal images of the femoral and tibial bone regions were generated by CTVol v2.0 software (Skyscan 1076; Skyscan, Luxembourg, Belgium).

### 2.6. Histopathology of the Articular Cartilage of the Knees

For articular cartilage histopathologic analysis, the knees were fixed in 10% neutral buffered formalin overnight and were decalcified in 10% PBS-buffered EDTA (Sigma Aldrich, St. Louis, MO, USA) at 37 °C for approximately 1 month until the bones softened. The knees were embedded in paraffin wax blocks. The paraffin blocks were cut using a microtome for dissection into 5-μm-thick sections (Leica, Wetzlar, Germany), and the sections were stained with safranin-O (Sigma Aldrich, USA). The stained sections were then mounted, and histopathological changes were observed. The Osteoarthritis Research Society International (OARSI) cartilage osteoarthritis (OA) grading scores were measured on a 0-to-24 scale by the index of grades with stages [21].

### 2.7. Enzyme-Linked Immunosorbent Assay 

The serum levels of bone morphogenetic protein 2 (BMP2) (DBP200, R&D System, MN, USA) and Insulin-like growth factor 1 (IGF1) were estimated using rat ELISA kits (MG100, R&D System, MN, USA).

### 2.8. Immunohistochemistry 

Decalcified sections of the proximal tibia were probed with primary antibodies against BMP2 (ab6285, Abcam, San Francisco, CA, USA), BMP4 (ab39973, Abcam, San Francisco, CA, USA), SMAD family member 4 (Smad4; sc-7966, Santa. Cruz Biotechnology, Santa Cruz, CA, USA), Wnt3a (#2391, Cell Signaling Technology, Inc, Danvers, MA, USA), and Cyclin D1 (A-12, Santa. Cruz Biotechnology, Santa Cruz, CA, USA) overnight at 4 °C. To examine the BMP2 and BMP4 signaling pathways, antibodies against BMP2, BMP4, and Smad4 were used. For examination of the Wnt3a signaling pathway, antibodies against Wnt3a and Cyclin D1 were used. Detection was performed using a goat anti-rabbit horseradish peroxidase-conjugated and 3′-, 3′-diaminobenzidine (DAB) cell and tissue staining kit (R&D Systems, Inc., Minneapolis, MN, USA). The immunoactivities were measured from five areas in three sections of each specimen by using a Zeiss Axioskop 2 plus microscope (Carl Zeiss, Gottingen, Germany). All images of each specimen were captured using a cool CCD camera (Media Cybernetics Inc., Rockville, MD, USA). Images were analyzed by manual counting and confirmed using NIH ImageJ (vision 1.53g, NIH, Bethesda, MD, USA) and Image-Pro Plus (ver. 7.0.1.658, Media Cybernetics, Rockville, MD, USA).

### 2.9. Statistical Analysis 

Statistical analyses were performed using SPSS version 17.0 (SPSS Inc., Chicago, IL, USA). All data were expressed as arithmetic mean values with standard deviation (mean ± SD) or with their standard error of the mean (mean ± SEM). The statistical significances of differences were calculated using one-way analysis of variance (ANOVA) followed by post hoc Tukey–Kramer Multiple Comparison test or two-way ANOVA followed by the Bonferroni post hoc test. Statistical tests were used for comparisons between groups, and statistical significance was set at *p* < 0.05, *p* < 0.01, and *p* < 0.001.

## 3. Results

### 3.1. Improvement in Bone Strength of the Right Femur after SW on the Left Femur or Left Tibia in OVX Rats

In order to test if SW on bone induces growth factors to prevent bone loss, the experimental design and application of SW to the selected positions of the left femur or the left tibia in an OVX rat model are shown in Figure 1. The bone strength test was performed to observe the bone strength of the right femur after SW applied on the left femur (SW(F)) or the left tibia (SW(T)) (Figure 1B,C; Table 1). The modulus of the right femur was significantly increased in the SW(F) group, and it was increased, although not significant, in the SW(T) group in comparison with the OP group (Table 1). In addition, the extension at peak load, handleability, and strain at break were significantly improved in the right femur of the SW(T) group in comparison with the OP group. However, although the results were not statistically significant, positive trends in the extension at peak load, handleability, and strain at break of both femurs were observed in the SW(F) group (Table 1). The results displayed a protective effect in bone strength in the SW(T) and SW(F) groups, and the improvement in bone strength was greater in the SW(T) group than the SW(F) group.

### 3.2. SW to the Left Tibia or Left Femur Reduced Bone Loss in OVX Rats 

The micro-CT images visualized the bone structures of the Sham, OP, SW(T), and SW(F) groups (Figure 2). The micro-CT images of the OP group revealed that ovariectomy induced loss of cancellous bone from the transverse and sagittal views compared with the Sham group. In the SW(T) and SW(F) groups, bone loss was reduced in both the right and left tibia (Figure 2; the volume of interest (VOI) region). The results observed that prevention of bone loss was obvious in the SW(T) group compared to the SW(F) group. Then, we further analyzed the VOI region for BMD, TbTh, BV/TV, and bone porosity.

### 3.3. Micro-CT Analysis of Bone Quality after SW Therapy in OVX Rats

In the SW(T) and SW(F) groups, the BMD, TbTh, BV/TV, and porosity of both tibias and femurs in the OVX rats were improved by comparison with the OP group (*p* < 0.05, *p* < 0.01, and *p* < 0.001) (Figure 2, and Table 2 for tibias and Table 3 for femurs). Furthermore, the SW(T) group exhibited significantly improved BMD, TbTh, BV/TV, and porosity of the right and left tibias and femurs compared to the SW(F) group (Table 2 for tibias and Table 3 for femurs). The results demonstrated that treatment administered in the SW(T) group had better protective effects than in the SW(F) group in terms of improving bone recovery of both lower limbs in OVX rats.

### 3.4. SW on Bone Stabilized BMP2 Expression in the Serum and May Have Induced Long-Range Effects to Protect the Extracellular Matrix of the Articular Cartilage in Both Lower Limbs of OVX Rat

BMP2 and IGF1 are important factors in bone and cartilage recovery. The serum levels of BMP2 and IGF1 were surveyed after SW to the left femur or the left tibia in OVX rats (Figure 3A). The level of serum BMP2 was significantly not reduced in the SW(T) and SW(F) groups in comparison with the Sham and OP groups, and the SW(T) group exhibited better results than the SW(F) group in terms of maintaining BMP2 in the serum (Figure 3A). However, the level of IGF1 was not significantly different when compared to the Sham, OP, and SW treatment groups. 

In the analysis of both knees, we observed that the extracellular matrix was decreased, a small area had uncalcified thickness and was slightly damaged in the OP group, and fibrous tissue formed on the surface of the articular cartilage of both knees by safranin-O staining (Figure 3B, arrow head). However, the damage level of articular cartilage was not significantly different by OARSI score (data not shown). The SW(T) and SW(F) groups exhibited improved articular cartilage recovery in both knees in comparison with the OP group (Figure 3B). Safranin-O staining showed that SW prevented the loss of cartilage matrix in the knees with a long range-effect following treatment. The results may suggest that SW maintained the level of serum BMP2 to protect the extracellular matrix proteins of the articular cartilage in both knees. 

### 3.5. SW Enhanced the Expressions of BMP2, BMP4, and Wnt3a Signaling in the Articular Cartilage of Osteoporotic Knees

In fact, in the protection of articular cartilage, not only BMP2 but also BMP4 are crucial for chondrocyte proliferation and matrix deposition. BMP2 and BMP4 belong to the transforming growth factor β (TGF-β) superfamily, which plays important roles in cartilage repair, development, and metabolism. Immunohistochemical analyses showed that BMP2, BMP4, and Smad4 were expressed in articular cartilage in the Sham, OP, SW(T), and SW(F) groups (Figure 4). The expressions of BMP2, BMP4, and Smad4 were obviously increased in the cartilage of both knees after SW on the left tibia or femur in comparison with the OP group (*p* < 0.001). 

Further, the BMP and Wnt signaling pathways play critical roles in cartilage homeostasis, proliferation, and differentiation. Wnt3a and Cyclin D1, which are the key factors in the Wnt signaling pathway, were surveyed in the articular cartilage after SW by immunohistochemical analysis and were found significantly (*p* < 0.001) decreased in the OP group and increased in the SW(T) and SW(F) groups (Figure 5). These results demonstrated that SW on the left tibia or femur induced long-range protective effects to modulate the key factors of BMP and Wnt signaling to articular cartilage repair in the osteoporotic knee.

## 4. Discussion

In the present study, we compared the effects of SW on the left femur or on the left tibia to treat OP disease in an OVX rat model. SW applied on the left tibia displayed better long-range improvement than on the left femur in terms of bone protection and against degeneration of articular cartilage in both knees in OVX rats. A specific growth factor, BMP2, which promotes bone formation and remodeling exhibited stabilized expression in the serum of OVX rats after SW treatment. In addition, BMP2, BMP4/Smad4, and Wnt3a/Cyclin D1 signaling key factors were stimulated to protect against degeneration of articular cartilage of the knees in OVX rats. Our current data suggested that SW on bone may induce the expression of BMP2 to have long-range protective effects for OP disease and osteoporotic arthritis.

SW increases the expressions of numerous bone formation growth factors and further influences the differentiation of bone marrow stromal cells towards osteoprogenitors [25]. More recent in vitro studies have reported that SW could stimulate osteoblasts and could increase periosteal cell viability directly [28]. This study displayed that SW significantly prevents bone loss or changes in bone turnover rates and stop OVX-induced damage on bone microarchitecture and bone matrix (Figure 2). The results indicated that, in our protocol, SW therapy might be useful for the treatment of osteopenia and OP of bone in both lower limbs. 

BMP2 is a significant growth factor in homeostasis of cartilage and bone in the human body [29,30]. Researchers have reported that an increasing serum BMP2 level promoted osteoporotic fracture-healing by activating the BMP/Smad signaling pathway in patients [31]. In addition, SW on bone increased the levels of serum and tissue BMP2 in the treatment of osteonecrosis of the femur head in patients and in animal studies to improve bone remodeling [32,33]. However, these studies demonstrated that SW had a local effect on bone healing [21,34,35]. Here, a hypothesis for this experiment is that SW on bone could induce the key factors from bone marrow to serum and affects bone recovery in OVX rats. In the current study, we observed that SW applied to the left tibia or left femur in OVX rats not only stabilized the serum level of BMP2 but also increased the expression of BMP2 in the articular cartilage of both knees, supporting the abovementioned hypothesis (Figure 3A and Figure 4). On the other hand, IGF1 is a critical factor in bone growth and skeletal maturation for the human body [36]. Lower serum IGF-1 levels are correlated with older postmenopausal women and increase the risk of bone fracture [37]. Therefore, in this experiment, the serum level of IGF1 would be affected by the time of induced OVX in rat. In Martina Böttner’s study, the OVX rat was created by bilateral ovariectomy, as in our study [38]. Their data indicated that the serum level of IGF1 was no different in the OVX group post-surgery at 12 weeks compared with intact group. The results were the same as in our study. This data explained why the serum level of IGF1 in OVX rat would not change post-surgery at 12 weeks and after SW treatment.

Following our protocol (0.25 mJ/mm^2^ with 1600 impulses in total), SW induced long-range effects to improve the BMD in both lower limbs and to protect the articular cartilage of both knees in OVX rats. In 2017, Lama reported a systemic effect of SW (0.33 mJ/mm^2^ with 1000 impulses) to the right hind leg using a protocol consisting of SW five times per week and total treatments for five weeks followed by sacrifice of the rats [27]. In this study, we compared the effects of SW applied on the left tibia or left femur and observed that SW on the left tibia led to better results than SW on the left femur for OVX rats. The differences between our study and that of Lama were as follows: (1) Our SW was applied at different locations, the left medial tibia or femur in our experiment compared to the antero-lateral side of the right thigh hind leg in Lama’s experiment. (2) The protocol of our experiment was a one-shot prevention treatment one week post-surgery. On the contrary, Lama’s study was a five times per week for five weeks treatment after 16 weeks post-surgery. (3) The articular cartilage of both knees of OVX rats were protected in our study. (4) We achieved long-range protective effects from this study compared to the systemic effect of Lama’s experiment. Finally, the results of our study and that of Lama suggested that SW could be an alternative method for clinical treatment of OP systemically. 

SW has been reported to exert a chondroprotective effect and to improve bone formation and bone healing [19,39]. This physical therapy is a noninvasive method that improves local BMD, callus endurance, and bone microarchitecture and reduces osteoporotic fractures [40,41]. Our previous study showed that SW improved subchondral bone remodeling in osteoporotic arthritis in a rat model [21]. It has been suggested that subchondral bone remodeling plays an important role in the progression of cartilage degenerative change. Hayami demonstrated that subchondral bone resorption is significantly associated with cartilage thinning and subchondral bone sclerosis in the OA knee [42]. In the present study, we found that subchondral bone change in OVX rats was directly correlated with the grade of articular cartilage degeneration (Figure 3B). Bone and cartilage growth signals, such as BMPs and Wnts, restored the BMD and bone microarchitecture in knees following SW and were associated with cartilage regeneration (Figure 4 and Figure 5). These results showed that changes in the underlying subchondral bone of both knees following SW can protect against OA progression.

There were some limitations in our study. Small animals were used in the experiments, and the results therefore are not directly applicable to humans after SW. The protocol of SW application was a significant limitation, and a standard procedure is required for further assessment of the method for treatment of human OP. The source of SW was another limitation; different sources of SW have differing energy transmissions to the bone and induce varying levels of biological effects. Variations exist among rats, and the amount of bone tissue is much smaller; therefore, the energy in SW is distributed differently, which may cause a difference in the magnitude of the biological effects. Taken together, our results are impressive, and this physical therapeutic technique appears to be promising for the treatment of OP.

## 5. Conclusions

The results of this study displayed that shockwave therapy improved bone strength and bone recovery in both femurs and tibias in osteoporotic rats. Further, SW on bone stimulated the expressions of serum BMP2 as well as induced BMP2, BMP4, Smad4, Wnt3a, and cyclinD1 in the articular cartilage of osteoporotic rat knees. Our results also demonstrated that SW on bone may have long-range protective effects to ameliorate OP disease and degeneration of knee cartilage in OVX rats simultaneously.

## Figures and Tables

**Figure 1 biomedicines-08-00614-f001:**
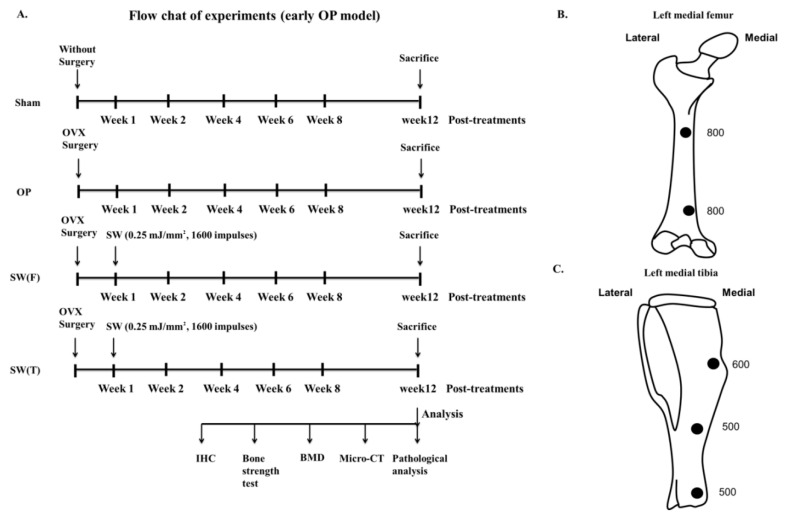
Study design: (**A**) the figure indicates the performance of surgery, treatment, sacrifice of rats, and analysis in each group. Shockwave (SW) application was focused (**B**) on the left medial femur (SW(F)) with 800 impulses or (**C**) on the left tibia (SW(T)) with 600, 500 and 500 impulses at 0.5 cm below the skin of the rat at each location selected (black circles). Osteoporosis is denoted OP. The ovariectomized is denoted OVX. Immunohistochemistry is denoted IHC. The bone mineral density is denoted BMD and micro computed tomography is denoted micro-CT.

**Figure 2 biomedicines-08-00614-f002:**
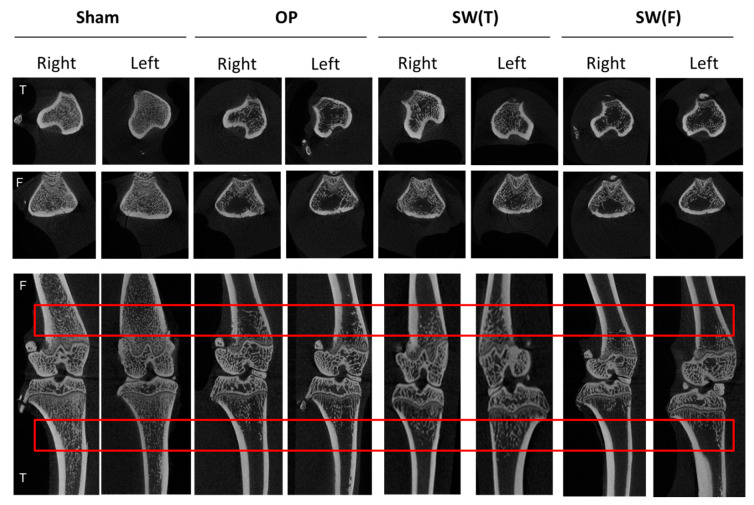
Micro-CT scans of different groups: the results are displayed on photomicrographs of the left and right knee in transverse (tibia and femur, up) and sagittal (femur and tibia, below) views obtained using micro-CT scanning. The volume of interest (VOI) region contained within the red line denotes the area of inspection. N = 6. The tibia is denoted T, and the femur is denoted F.

**Figure 3 biomedicines-08-00614-f003:**
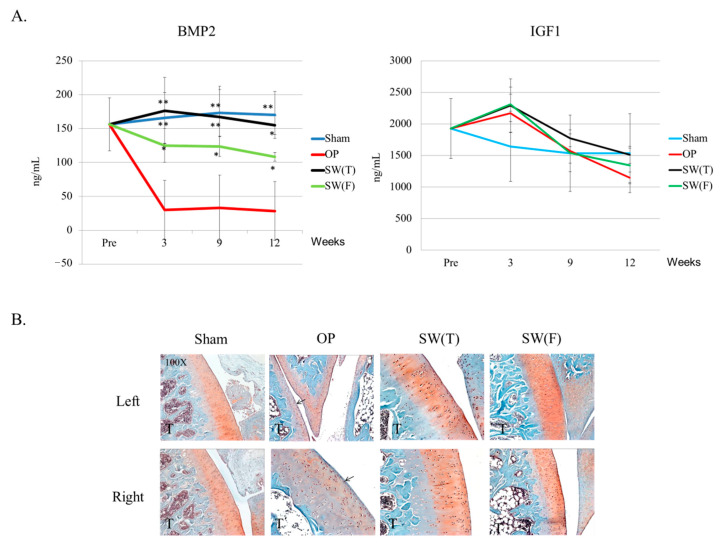
Enzyme-linked immunosorbent assay (ELISA) and protection of the articular cartilage of the knee: (**A**) the serum bone morphogenetic protein 2 (BMP2) and Insulin-like growth factor 1 (IGF1) levels were surveyed by ELISA. Values are the mean ± SD. * *p* < 0.05 and ** *p* < 0.01 versus the OP group. N = 6. (**B**) The images of the left and right articular cartilages of the knees showed changes in the articular cartilage in the different groups following safranin-O staining (×100 magnification). The arrow heads indicate fibrous tissues. N = 6. The tibia is denoted T.

**Figure 4 biomedicines-08-00614-f004:**
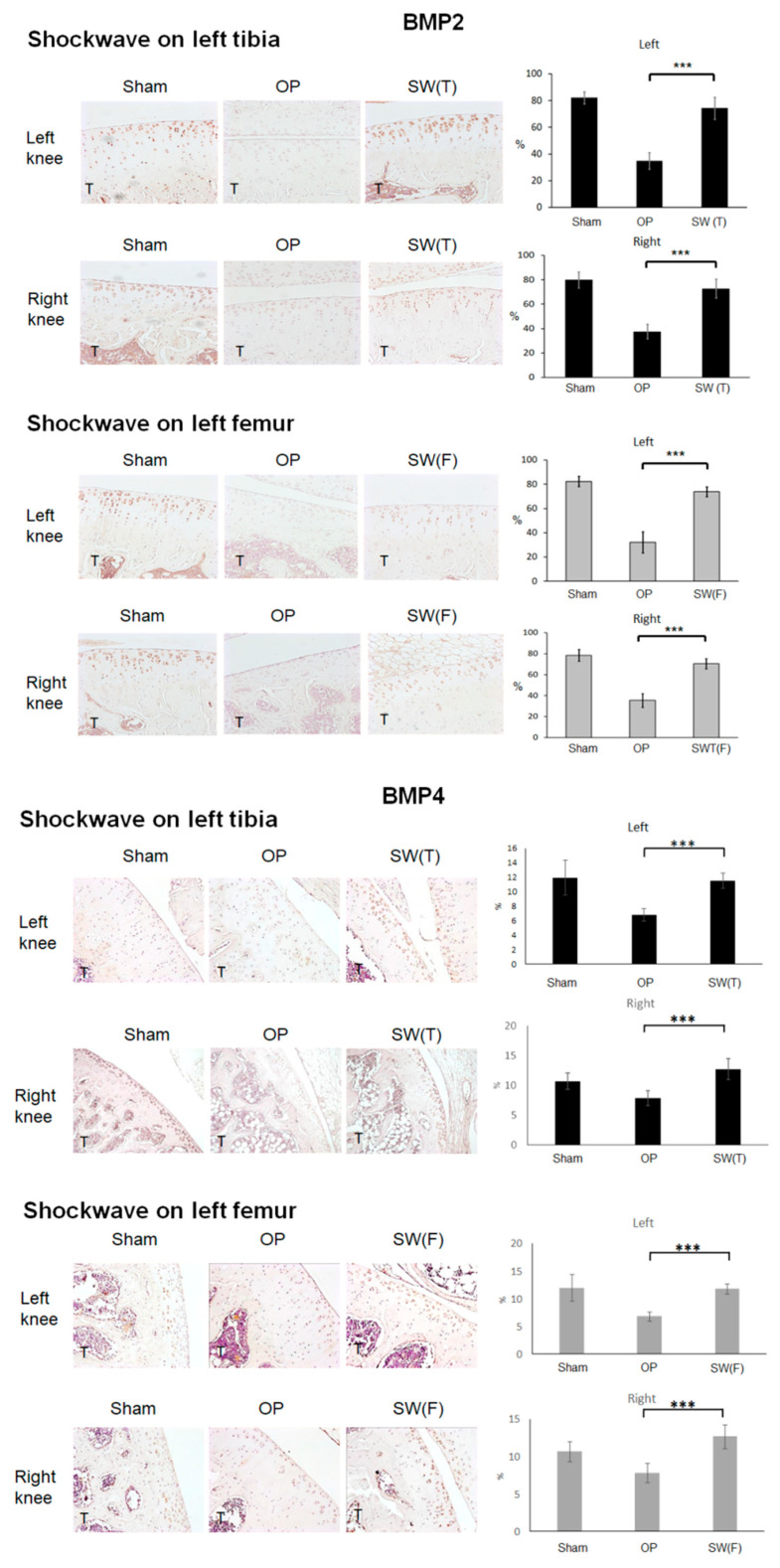
Shockwave therapy induced long-range effects on the expressions of BMP2, BMP4, and SMAD family member 4 (Smad4) in the articular cartilage (×100 magnification) of the tibia of the right and left knees in the Sham, OP, SW(T), and SW(F) groups. T = tibia. *** *p* < 0.001. Values are the mean ± SEM.

**Figure 5 biomedicines-08-00614-f005:**
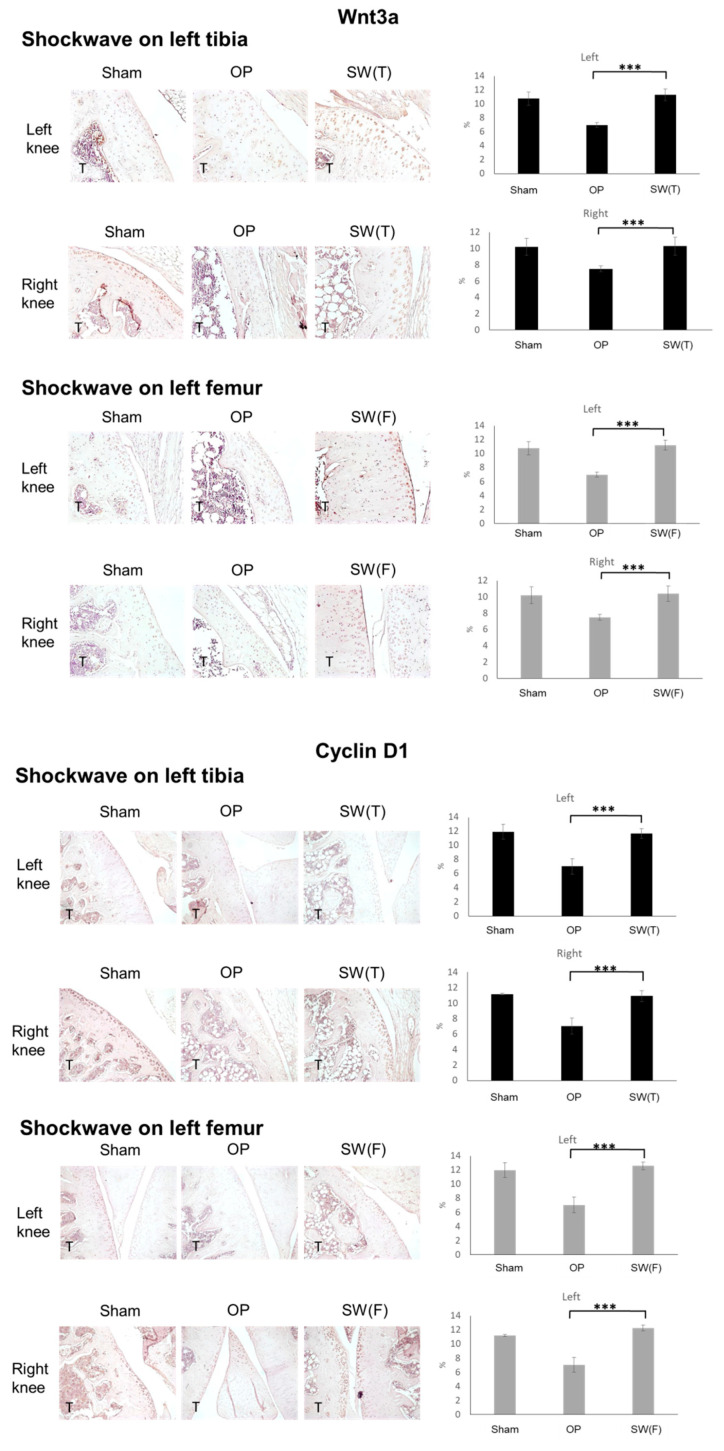
Long-range effects of shockwave therapy on the expressions of Wnt3a and Cyclin D1 in the articular cartilage (×100 magnification) of the tibia of the right and left knees in the Sham, OP, SW(T), and SW(F) groups. T = tibia. N = 6. *** *p* < 0.001. Values are the mean ± SEM.

**Table 1 biomedicines-08-00614-t001:** The right femur bone was harvested for bone strength tests with modulus, extension at peak load, handleability, and strain at break.

	Modulus			
Legs	Sham	OP	SW(F) ^1^	SW(T) ^1^
Right	382.45 ± 73.33 *	279.55 ± 37.91	412.81 ± 87.45 *	314.15 ± 30.05
Extension at peak load				
Legs	Sham	OP	SW(F)	SW(T)
Right	0.82 ± 0.15	0.71 ± 0.10	0.88 ± 0.13	0.96 ± 0.14 *
Handleability				
Legs	Sham	OP	SW(F)	SW(T)
Right	12.08 ± 3.56	11.85 ± 2.53	14.93 ± 1.59	16.99 ± 3.04 *
Strain at break				
Legs	Sham	OP	SW(F)	SW(T)
Right	10.86 ± 2.03	9.45 ± 1.34	11.88 ± 1.33	12.79 ± 1.86 *

^1.^ SW(T) and SW(F) indicate shockwave applied on the tibia or femur. Abbreviation: T = tibia; F = femur. Values are mean ± SEM, N = 6. * *p* < 0.05 versus osteoporosis (OP).

**Table 2 biomedicines-08-00614-t002:** Both VOI regions of the right and left tibias for bone mineral density (BMD), trabecular thickness (TbTh), bone volume/tissue volume (BV/TV), and porosity of ovariectomized (OVX) rats in the Sham, OP, SW(F), and SW(T) groups by micro-CT analysis.

Bone Mineral Density (BMD; g/mm^3^)				
Legs	Sham	OP	SW(F) ^a^	SW(T) ^a^
Right	0.46 ± 0.02	0.23 ± 0.01	0.27 ± 0.01 **	0.30 ± 0.01 **
Left	0.46 ± 0.03	0.23 ± 0.01	0.27 ± 0.01 **	0.31 ± 0.01 **
Trabecular Thickness (TbTh; mm)				
Legs	Sham	OP	SW(F)	SW(T)
Right	0.09 ± 0.01	0.09 ± 0.01	0.12 ± 0.01 ***	0.12 ± 0.01 ***
Left	0.09 ± 0.01	0.10 ± 0.01	0.11 ± 0.01 **	0.13 ± 0.01 **
Bone Volume/Tissue Volume (BV/TV; %)				
Legs	Sham	OP	SW(F)	SW(T)
Right	45.60 ± 0.04	27.84 ± 0.01	28.82 ± 0.05	31.61 ± 0.03 *
Left	45.64 ± 0.06	27.16 ± 0.02	29.78 ± 0.03	32.89 ± 0.04 *
Porosity (%)				
Legs	Sham	OP	SW(F)	SW(T)
Right	54.17 ± 3.15	72.16 ± 1.35	71.33 ± 4.76	68.15 ± 2.82 **
Left	53.83 ± 4.53	72.84 ± 1.57	70.06 ± 3.10	67.38 ± 3.89 **

^a^ SW(T) and SW(F) indicate shockwave applied on the tibia or the femur. Abbreviation: T = tibia; F = femur. Values are mean ± SEM, N = 6. * *p* < 0.05, ** *p* < 0.01, and *** *p* < 0.001 versus OP.

**Table 3 biomedicines-08-00614-t003:** Both VOI regions of the right and left femurs for BMD, TbTh, BV/TV, and porosity of OVX rats in the Sham, OP, SW(F), and SW(T) groups by micro-CT analysis.

Bone Mineral Density (BMD; g/mm^3^)				
Legs	Sham	OP	SW(F) ^a^	SW(T) ^a^
Right	0.42 ± 0.02 ***	0.19 ± 0.01	0.18 ± 0.01	0.23 ± 0.01 *
Left	0.41 ± 0.02 ***	0.18 ± 0.01	0.18 ± 0.01	0.23 ± 0.01 *
Trabecular Thickness (TbTh; mm)				
Legs	Sham	OP	SW(F)	SW(T)
Right	0.1 ± 0.01	0.1 ± 0.01	0.11 ± 0.01	0.1 ± 0.01
Left	0.1 ± 0.01	0.1 ± 0.01	0.1 ± 0.01	0.1 ± 0.01
Bone Volume/Tissue Volume (BV/TV; %)				
Legs	Sham	OP	SW(F)	SW(T)
Right	40.83 ± 1.67 ***	19.33 ± 0.72	18.5 ± 0.62	23.97 ± 1.22 *
Left	39.77 ± 1.60 ***	19.03 ± 1.15	19.34 ± 0.77	23.38 ± 0.89 *
Porosity (%)				
Legs	Sham	OP	SW(F)	SW(T)
Right	59.16 ± 1.67 ***	80.67 ± 0.72	81.5 ± 0.62	76.03 ± 1.22 *
Left	60.22 ± 1.60 ***	80.97 ± 1.15	80.66 ± 0.77	75.6 ± 0.69 *

^a^ The SW(T) and SW(F) are indicated shockwave applied on the tibia or the femur. Abbreviation: T = tibia, F = femur. Values are mean ± SEM, N = 6. The * *p* < 0.05, *** *p* <0.001 versus OP.

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
