# Peer review of "Shockwave Therapy Modulates the Expression of BMP2 for Prevention of Bone and Cartilage Loss in the Lower Limbs of Postmenopausal Osteoporosis Rat Model"

_biomedicines, 2020, doi:10.3390/biomedicines8120614_

Round 1
Reviewer 1 Report
The authors studied the effect of shockwave (SW) therapy on bone and cartilage in an osteoporosis rat model. They found that SW improved cartilage and increased markers of cartilage formation and healing. Only two minor comments/questions?
1) Did the authors also look at the effects of SW on synovial membrane and fluid? Any changes in levels of proinflammatory markers?
2) The tables show “mean ± SEM”. Please reduce the decimal places after the comma for SEM. For example, table 2: “0.46±0.019“ to “0.46±0.02“
Author Response
Review 1:
In this manuscript, all authors agree to revise and submit the revision to Biomedicines. The authors appreciated the suggestions of editor and reviewers. We were point-by-point to response the comments as below. In this major revision, we would like to thank the reviewer1 for careful and thorough reading of this manuscript and for the thoughtful comments and constructive suggestions.
Comments and Suggestions for Authors
The authors studied the effect of shockwave (SW) therapy on bone and cartilage in an osteoporosis rat model. They found that SW improved cartilage and increased markers of cartilage formation and healing. Only two minor comments/questions?
1) Did the authors also look at the effects of SW on synovial membrane and fluid? Any changes in levels of proinflammatory markers?
Response: Thanks reviewer’s suggestion. The synovium was checked by HE staining but there was no obviously pathological inflammation in OP group by compared with Sham group as the below Figure1 (Showed in PDF file). Therefore, we did measure the proinflammatory markers. We did not survey the synovial fluid because it was very low-volume to analyze in this study. We already proceed the further study, and we maybe to survey the levels of proinflammatory markers such as IL1β and IL6 in the experiment.
2) The tables show “mean ± SEM”. Please reduce the decimal places after the comma for SEM. For example, table 2: “0.46±0.019“ to “0.46±0.02“
Response: Thanks reviewer’s suggestion. We reduced the decimal places after comma for SEM in the Tables.

Reviewer 2 Report
The animal model used in this work may be compared to a previous one recently described (Bone. 2018 May 110: 1–10. doi:10.1016/j.bone.2018.01.019). That should be cited in the text and commented by the authors.
Please provide an explanation of your experimental study design. It is important to justify the choice of the SW doses, the area of application and how you decided the interval days between the different treatments. Also, an effective demonstration of the SW-induced dose-dependent stimulatory effect is totally lacking in this work.
In this paper, BMD is mainly measured by micro-CT analysis, whereas bone and cartilage recovery have been evaluated by ELISA and immunohistochemistry.
On my point of view, a more exhaustive and appropriated morphology of bone and cartilage tissue microarchitecture should be performed using electron microscopy.
ELISA test was employed for surveying the serum levels of BMP2 and IGF1.
The amount of IGF1 was not significantly different among the treatment groups, please provide a possible explanation and comment this result in the discussion.
No clear evidence for SW-induced tissue repair and differentiation can be demonstrated by serum BMP2 levels, whereas further markers should be examined. For example, the osteogenic potential can be measured by an alkaline phosphatase assay, that can be indicative for bone activity and differentiation. In addition, even though I agree that BMP plays a key role in cell proliferation and differentiation, being important prerequisites for tissue regeneration, no mitogenic potential of SW treatment was directly ascertained in this research.
Immunochemistry shows here that SW exposure increases the expressions of BMP2, BMP4 and Smad4, corroborating the idea that such treatments may protect from bone and cartilage degeneration. Those arguments could be more convincing if supported by molecular analysis of rat tissue samples through RT-PCR, able to evaluate cartilage and bone status and turnover markers, including SOX9, Collagens I, II, III and alkaline phosphatase.
Author Response
Review 2:
In this manuscript, all authors agree to revise and submit the revision to Biomedicines. The authors appreciated the suggestions of editor and reviewers. We were point-by-point to response the comments as below. In this major revision, we would like to thank the reviewer2 for careful and thorough reading of this manuscript and for the thoughtful comments and constructive suggestions.
Comments and Suggestions for Authors
The animal model used in this work may be compared to a previous one recently described (Bone. 2018 May 110: 1–10. doi:10.1016/j.bone.2018.01.019). That should be cited in the text and commented by the authors.
Response: Thanks reviewer’s suggestion. We cited the reference in the Introduction. (Introduction line 41).
Please provide an explanation of your experimental study design. It is important to justify the choice of the SW doses, the area of application and how you decided the interval days between the different treatments. Also, an effective demonstration of the SW-induced dose-dependent stimulatory effect is totally lacking in this work.
Response: Thank you very much for the reviewer’s suggestion. We know that SW can induce local effects in the region of bone loss but rare evidences demonstrate the systemic effect in bone regeneration. According to the experience of our studies in the osteoarthritis (OA) treatment, we found SW on medial site was better than lateral site for OA to stimulate bone remodeling. Therefore, in this study, we try to apply the SW on different medial site of femur or tibia to stimulate bone regeneration. On other idea, we hypothesize SW could stimulate bone marrow to release growth factors to serum and improve bone recovery systemically. This phenomenon was observed in the study, but lack the data of micro-CT analysis in spine to elucidate the systemic effect of SW. Therefore, we are proceeding the new experiment to demonstrated the systemic effect of SW and will also display the dose-dependent effect in the future.
In this paper, BMD is mainly measured by micro-CT analysis, whereas bone and cartilage recovery have been evaluated by ELISA and immunohistochemistry. On my point of view, a more exhaustive and appropriated morphology of bone and cartilage tissue microarchitecture should be performed using electron microscopy.
Response: Thanks reviewer’s suggestion. The method of electron microscopy is a high technology to analyze the bone mineralization. We will use this high technology to observe the detail changes in the bone and articular cartilage in the next osteoporosis study.
ELISA test was employed for surveying the serum levels of BMP2 and IGF1. The amount of IGF1 was not significantly different among the treatment groups, please provide a possible explanation and comment this result in the discussion.
Response: Thanks reviewer’s suggestion. We revised the comments of IGF1 in the section of Discussion as following (Line 291): IGF1 is a impotent factor in bone growth, and skeletal maturation for human body (1). Lower serum IGF-1 levels are correlated with the elder postmenopausal women and increasing the risk of bone fracture (2). Therefore, in the experiment, the serum level of IGF1 would be affected by the time of induced OVX in rat. In the Dr. Martina Bo¨ttner’s study, the OVX rat was created by the bilateral ovariectomy as the same with our study (3). Their data indicated that the serum level of IGF1 was no difference in OVX group post-surgery at 12 weeks by compared with Intact group. The results was the same with our study. This data explanted why the serum level of IGF1 in the OVX rat would not change post-surgery at 12 weeks and after SW treatment. Our unpublished data displayed that serum IGF1 was decreased in the OVX rat by inducing after 24 weeks. Then the serum IGF1 was improved by SW as the below figure (Showed in PDF file).
Reference:
- Giustina, G. Mazziotti, and E. Canalis, “Growth hormone, insulin-like growth factors, and the skeleton,” Endocrine Reviews, vol. 29, no. 5, pp. 535–559, 2008.
- Kanazawa, T. Yamaguchi, M. Yamamoto, M. Yamauchi, S. Yano, and T. Sugimoto, “Serum insulin-like growth factor-I level is associated with the presence of vertebral fractures in postmenopausal women with type 2 diabetes mellitus,” Osteoporosis International, vol. 18, no. 12, pp. 1675–1681
- M Böttner, J Christoffel, W Wuttke, “Effects of long-term treatment with 8-prenylnaringenin and oral estradiol on the GH-IGF-1 axis and lipid metabolism in rats.” J Endocrinol. 2008 Aug;198(2):395-401.
No clear evidence for SW-induced tissue repair and differentiation can be demonstrated by serum BMP2 levels, whereas further markers should be examined. For example, the osteogenic potential can be measured by an alkaline phosphatase assay, that can be indicative for bone activity and differentiation. In addition, even though I agree that BMP plays a key role in cell proliferation and differentiation, being important prerequisites for tissue regeneration, no mitogenic potential of SW treatment was directly ascertained in this research.
Response: Thank you for this suggestion. It would have been interesting to explore this aspect in SW treatment. This is our goal to demonstrate that SW induced serum BMP2 to promote tissue regeneration directly in next experiment.
Immunochemistry shows here that SW exposure increases the expressions of BMP2, BMP4 and Smad4, corroborating the idea that such treatments may protect from bone and cartilage degeneration. Those arguments could be more convincing if supported by molecular analysis of rat tissue samples through RT-PCR, able to evaluate cartilage and bone status and turnover markers, including SOX9, Collagens I, II, III and alkaline phosphatase.
Response: Thank you for this suggestion. In the study, we focused on the immunochemical staining. Indeed, further molecular analysis such as RT-PCR or Western blot could help us to have more convincing results. We thanks the reviewer’s courage and suggestion, and we will do our best in the next study from the limiting budget.
